# Urine Protein to Creatinine Ratio for the Assessment of Bevacizumab-Associated Proteinuria in Patients with Gynecologic Cancers: A Diagnostic and Quality Improvement Study

**DOI:** 10.3390/diagnostics14171852

**Published:** 2024-08-24

**Authors:** Kuan-Ju Huang, Wen-Chun Chang, Chi-Hau Chen, Wei-Chen Lin, William Wei-Lin Pan, Hao-I. Hsieh, Yu-Hsiung Hsieh, Lin-Hung Wei, Bor-Ching Sheu

**Affiliations:** 1Department of Obstetrics and Gynecology, National Taiwan University Hospital Yunlin Branch, Yunlin 640203, Taiwan; 2Graduate Institute of Clinical Medicine, National Taiwan University College of Medicine, Taipei 100, Taiwan; 3Department of Obstetrics and Gynecology, National Taiwan University Hospital, National Taiwan University College of Medicine, Taipei 100, Taiwan; 4Xing Kang Clinic, Hualien 970, Taiwan; 5Department of Medical Education, National Taiwan University Hospital, National Taiwan University College of Medicine, Taipei 100, Taiwan

**Keywords:** urine protein to creatinine ratio, bevacizumab, angiogenesis inhibitors, antineoplastic agents, proteinuria, albuminuria

## Abstract

Proteinuria is a common adverse event arising from treatment with bevacizumab, requiring diagnostic testing via 24-h urine collection. However, this method is cumbersome. We assessed urine screenings in gynecologic cancer patients from February 2021 to May 2022. Along with a simple urine dipstick (UD), the urine microalbumin, total protein, and creatinine were measured and calculated as the urine albumin to creatinine ratio (UACR) and the urine protein to creatinine ratio (UPCR), which were further adjusted through the Modification of Diet in Renal Disease and Chronic Kidney Disease Epidemiology Collaboration equations to be estimated and correlated with 24-h urine total protein content. The incremental cost-effectiveness ratio was used for cost analysis. There were 129 urine samples from 36 patients. The sensitivity and specificity for the UACR were 0.56 and 0.97, and for the UPCR, 0.71 and 0.88, respectively. The 24-h TP correlated strongly with the UACR (r = 0.75; *p* < 0.001) and UPCR (r = 0.79; *p* < 0.001) and fair for the simple UD (r = 0.35; *p* < 0.001). The UPCR saves one unnecessary 24-h urine test for less than a dollar compared to a simple UD. The results indicate that using the UPCR could enhance diagnostic accuracy, lower costs, and reduce unnecessary 24-h urine sampling.

## 1. Introduction

Bevacizumab has been approved by the U.S. Food and Drug Administration (FDA) for the treatment of malignant neoplasms, particularly as first-line adjuvant therapy or salvage treatment in gynecologic cancer patients [1,2,3,4]. Severe proteinuria is a rare adverse effect of bevacizumab treatment [5]. The incidence is between 11.9 and 38.3% for all-grade proteinuria and 0.7 and6.5% for high-grade proteinuria, defined by the total urine protein content > 3.5 g/day or a 3+ and above result in the urine dipstick test (Appendix A) [5,6,7,8]. Currently, there is a lack of evidence-based guidelines for treating proteinuria in patients receiving VEGF-targeted agents. Baseline and periodic urinalysis, using the spot simple urine dipstick test (UD), are recommended for screening, and a further diagnostic 24-h urine collection is required for those who have a 2+ or higher result [5,8,9,10]. Generally, if protein excretion exceeds 2 g per day, a temporary suspension of bevacizumab treatment is recommended [8]. Additionally, permanent discontinuation of bevacizumab should be considered for patients who develop nephrotic syndrome [5]. However, the accuracy of a simple UD for detecting proteinuria depends on the patient’s activity, position, medical condition, and the quality of the sample. Many institutions skip routine simple UD before each bevacizumab cycle, often opting for the more time-consuming and cumbersome, yet accurate, 24-h urine test [11]. The spot urine (micro)albumin to creatinine ratio (UACR), measuring the levels of micro amounts of albumin compared to the amount of creatinine in the urine, serves as an early marker for kidney damage and correlates with 24-h quantitative proteinuria in glomerular diseases [12,13]. However, the mechanism behind bevacizumab-associated proteinuria is unclear, and several reports revealed histologic changes in both glomerular and non-glomerular injuries [5]. Conversely, the spot urine protein to creatinine ratio (UPCR) provides a broader picture of kidney damage and can be used to assess the severity of kidney disease. Though there has been post-marketing data submitted to the FDA, the validity of the UPCR for the estimation of 24-h urine total protein content was not comparable to those used in glomerular diseases and was negated in bevacizumab-associated proteinuria [8,14]. On the contrary, the correlation between the UPCR and the 24-h urine total protein content in another report was higher than expected [6]. The objective of this study was to assess the relationship between the severity of proteinuria identified by a spot simple urine dipstick test (UD), spot urine albumin to creatinine ratio dipstick (UACRD) test, spot urine microalbumin to creatinine ratio (UACR) test, spot urine total protein to creatinine ratio (UPCR) test, and 24-h urine total protein (TP) content in gynecologic cancer patients receiving bevacizumab.

## 2. Materials and Methods

Any gynecologic cancer patients receiving bevacizumab, regardless of the use of a front-line adjuvant therapy or a salvage treatment and having a 2+ or higher simple UD or UACRD result, met the inclusion criteria (Figure 1). All urine samples were collected before the index test. Patients who had incomplete urine sampling to calculate the total urine protein content were excluded. All patients’ medical records were retrospectively collected, which included age at diagnosis, body mass index (BMI), and underlying medical diseases, including diabetes mellitus (DM), hypertension (HTN), active renal dysfunction, and liver dysfunction. The use of nephrotoxic agents during bevacizumab treatment, including nonsteroidal anti-inflammatory drugs, aminoglycosides, and associated chemotherapy, was also recorded. Active renal dysfunction refers to renal conditions with acute kidney injury defined by the Kidney Disease: Improving Global Outcomes (KDIGO) criteria or chronic kidney disease. The original Modification of Diet in Renal Disease (MDRD) Equation was used for the estimated glomerular filtration rate (eGFR). Active liver dysfunction was defined by elevated liver tests beyond a normal range.

Spot urine and blood samples were obtained from each patient before each cycle of bevacizumab. In all cases, the simple UDs were obtained during normal waking hours in the daytime, and no specimens were the first voided sample. In our institution, the semiquantitative UACRD test is also available simultaneously to screen all patients for early detection of diabetic nephropathy, in which the urine albumin level was measured. Both the simple UD and UACRD tests give graded results. In the simple UD test results, 1+, 2+, 3+, and 4+ results represent urine protein content of 0.3 g/L, 1 g/L, 3 g/L, and 10 g/L, respectively (Appendix A). In the UACRD test results, 1+ and 2+ represent an albumin-to-creatinine ratio of 30–300 mg/g and ≥300 mg/g, respectively. In patients with a result of grade 2+ or higher from either the simple UD or UACRD screening tests, additional urine samples for the microalbumin, total protein, and creatinine concentration were collected, and again after the 24-h urine samples became available. Typically, patients were asked to undergo 24-h urine sampling during hospitalization. However, due to the pandemic, some patients collect their samples in an outpatient setting. For the estimation of 24-h total protein content, several models by multiplying the UACR by the expected 24-h creatinine generation using the MDRD equation (UACR-MDRD) and the Chronic Kidney Disease Epidemiology Collaboration equation (eAER-CKD-EPI) have been developed [15,16]. Similarly, the UPCR and estimated urine protein excretion using the MDRD equation (UPCR-MDRD) and the CKD-EPI equation (ePER-CKD-EPI) was adopted. The severity of proteinuria was defined by the National Cancer Institute’s Common Terminology Criteria for Adverse Events version 5.0. Based on the previous study, the cumulative bevacizumab dose for the risk of severe proteinuria was determined to be 4530 g [7]. A cost assessment was performed using institution-specific cost data, in which the costs for a simple UD test, UPCR, and 24-h urine total protein test are USD 3.25, 3.47, and 1.73, respectively.

Before recruitment, a sample size estimation was conducted. Assuming the prevalence of disease was 0.70, the specified precision was 0.1, and the confidence interval (CI) was 95%, it was estimated that 116 records were needed for a test with an expected sensitivity and specificity of 0.90 and 0.90, respectively. The correlation between the simple UD, UACRD, UACR, UACR-MDRD, eAER-CKD-EPI, UPCR, UPCR-MDRD, and ePER-CKD-EPI tests and the 24-h total protein content was assessed using a Pearson’s correlation coefficient (r) analysis. The strength of the linear relationship was considered weak, fair, moderate, and strong if r < 0.3, 0.3 ≤ r < 0.5, 0.5 ≤ r < 0.7, and r ≥ 0.7, respectively. To determine the cut-off value and diagnostic performance of these tests for a 24-h urine total protein content ≥ 2 g, a receiver operating characteristic (ROC) curve was used. The costs of each test were calculated by the sum of the cost of one screening test multiplied by the entire cohort and the cost of the 24-h urine protein level multiplied by the number of patients who had positive results in the screening test. The estimated cost-effectiveness for simple UD and UPCR screening tests was evaluated via the incremental cost-effectiveness ratio (ICER) based on the prevalence of proteinuria reported in the literature [5,6,7,8]. A *p*-value of 0.05 was used as the threshold for statistical significance. All statistical analyses were performed using standard software (SPSS version 23; SPSS Inc., Chicago, IL, USA).

## 3. Results

### 3.1. Patient Baseline Characteristics

From February 2021 to May 2022 at the National Taiwan University Hospital, 129 samples from 36 patients were recorded (Table 1). The mean age was 63.36 ± 11.48 years. Six (16.67%) patients had DM, and fourteen (38.89%) patients had HTN. The majority of patients (88.89%) had malignancies involving the ovary, fallopian tube, or peritoneum. The mean bevacizumab dosage was 162.69 ± 109.40 mg/kg, and 15 patients (41.67%) were under concomitant chemotherapy. Among the 129 samples, the 24-h urine protein level was ≥2 g in 25 (19.38%) of them. The mean percentage of urine microalbumin to total protein content was 59.33 ± 18.94%. Forty samples (31.01%) were completed in an outpatient setting. Two (1.55%) samples from two patients showed stage I acute kidney injuries within 28 days of the bevacizumab treatments. The total urine protein levels were 3.43 g/day in one sample and 0.71 g/day in the other. One patient, whose total urine protein level was 0.25 g/day, died from pneumonia.

### 3.2. Correlations between Various Urine Screening Tests and 24-h Urine Protein Content

Table 2 summarizes the correlation (r) between the various urine screening tests and the 24-h urine total protein content. The UACRD was the least correlated with the 24-h urine total protein content (r = 0.1, 95% CI −0.12 to 0.32; *p* = 0.27). The simple UD had a fair correlation (r = 0.35, 95% CI 0.15 to 0.53; *p* < 0.01). In contrast, the UACR-MDRD, UACR, eAER-CKD-EPI, UPCR-MDRD, UPCR, and ePER-CKD-EPI had strong correlations (r = 0.74, 0.75, 0.78, 0.78, 0.79, and 0.82, respectively, all *p* < 0.001) (Figure 2).

### 3.3. Cost-Effectiveness of the Various Urine Tests

Considering the availability, accessibility, and convenience in clinical settings, we assessed the diagnostic performance of the UACRD, simple UD, UPCR, and ePER-CKD-EPI tests. For simple UD tests, the cut-off of a 2+ result was used, which is consistent with their use in most clinical scenarios. For the UACRD, the cut-off of a 2+ result was used to match for ≥grade 2 proteinuria (Appendix A). The Youden index for the UPCR was 1.83 (UPCR 1.83), and for the ePER-CKD-EPI, it was 2.06 (ePER 2.06), which were then used for calculating the diagnostic performances for the UPCR at the cut-off value of 1.83 (UPCR 1.83), the cut-off value of 2.00 (UPCR 2.00), and ePER at the cut-off value of 2.06 (Table 3). In brief, the UACRD had the highest sensitivity (0.92), followed by the simple UD (0.88), UPCR 1.83 (0.83), and UPCR 2.00 (0.71). The UPCR 2.00 had the highest specificity (0.88). The positive predictive values for the UACRD, simple UD, and UPCR were low. On the contrary, the negative predictive value for the UPCR 1.83 was 0.95, and 0.93 for the UPCR 2.00. Figure 3 shows the ROC curve for these tests. The area under the curve was highest for the ePER-CKD-EPI (0.92) and the UPCR (0.91), followed by the simple UD (0.60) and the UACR tests (0.54).

A stepwise proteinuria assessment test was defined as primary screening using either the ePER, UPCR, or simple UD, followed by a diagnostic test for 24-h total protein content for those who had positive results in the screening. The total cost per person was shown to be the lowest for the ePER 2.06 (USD 3.80), followed by the UPCR 2.00 (USD 3.88) (Table 3). Overall, compared to the simple UD, it costs less than one US dollar for UPCR tests to reduce one unnecessary, 24-h urine sampling in patients at risk for severe proteinuria. The savings could be greater if the prevalence of proteinuria is observed to be low due to fewer false positive results from screening tests (Appendix A). Three records exceeded the nephrotic range proteinuria (24-h urine protein content ≥ 3.5 g). The UPCR 2.00 had 100% sensitivity for this proteinuria level.

## 4. Discussion

The application of a UPCR for the estimation of 24-h total urine protein content was developed decades ago and was widely used with success in various conditions, including diabetic nephropathy, HTN, and glomerular disease (r = 0.96–0.97) [12,13,17]. It has been used to evaluate proteinuria of any grade in gynecologic cancer patients receiving bevacizumab [2,3,18,19]. However, post-marketing data showed that the correlation between the UPCR and the 24-h urine total protein content was low (r = 0.39), and the details of this study were limited [8]. Currently, the simple UD test is the standard screening test, which has high sensitivity (98.9%) for macroalbuminuria (>0.3 g/day) when the result is ≥1+ [20]. However, the simple UD test is relatively insensitive to non-albumin proteins, and the results may mask proteinuria caused by multi-factorial pathophysiology, as in the case of bevacizumab-associated proteinuria. Generally, a glomerular origin was considered to be the dominant etiology when the ratio of urine albumin to total protein concentration was greater than 40% [21]. The mean percentage of urine albumin to total protein in the current study was 59.33 ± 18.94%. The high percentage of urine albumin to total protein and high sensitivity of the simple UD test for detecting macroalbuminuria suggested that using the simple UD test to screen gynecologic cancer patients at risk of bevacizumab-associated proteinuria might be beneficial. Despite its strong correlation with the 24-h total urine protein content, the measurement of total protein may better predict proteinuria, considering the various percentages of albumin excretion observed in the current study. In addition, since microalbumin is a component in the UACR and is used for the early detection of microproteinuria in diabetic patients in which a highly sensitive test is required, it may give a large number of false positive results in patients receiving bevacizumab in which low-grade proteinuria is acceptable.

There is a lack of evidence for the optimal management of bevacizumab-associated proteinuria. It is recommended that if protein excretion is 2 g/day or more, bevacizumab should be temporarily withheld [8,10,11,22]. When protein excretion is in the nephrotic range, permanent discontinuation of bevacizumab is considered [3,10,19,22]. However, accurately completing a 24-h urine total protein test is difficult and costly, especially in areas with limited medical accessibility. This screening test could not only cause physiologic loading to both patients and medical staff but also an additional psychological burden on these cancer survivors. Most studies refer to the CTCAE for the assessment of proteinuria severity and use either the UPCR, simple UD, or 24-h urine total protein content as a guide for making decisions [2,4,7,10,18,23]. The prevalence of grade 1 to 2 proteinuria was reported as 4% in the Western population and up to 40% in the Asian population. The prevalence of grade 3 to 4 proteinuria is between 1 and 8.5% and 0 and 12.6% for non-Asian and Asian populations, respectively [2,4,5,6,7,16,20,24,25,26,27]. Normally, when the dipstick result is 2+ or more, a quantitative urine protein test follows [9]. However, a screening test with a high sensitivity used in a low-prevalence disease would result in many false positives. Many patients, therefore, need a second diagnostic test or have to postpone their bevacizumab treatments, which could be altered by medical accessibility, health equity, and psychosocial interference.

The correlation with the 24-h urine protein content was highest for the UPCR models (r = 0.82, 95% CI 0.75 to 0.88; *p* < 0.001 for the ePER-CKD-EPI, and r = 0.79, 95% CI 0.71 to 0.85; *p* < 0.001 for the UPCR), followed by the UACR models. The correlation between the simple UD test and the 24-h urine total protein content was fair (r = 0.35, 95% CI 0.15 to 0.53; *p* < 0.001) but weak for the UACRD test (r = 0.1, 95% CI −0.12 to 0.32; *p* = 0.31). Generally, the CKD-EPI model, adjusted for age, gender, ethnicity, and weight, had a better correlation efficiency than the original UACR and UPCR, followed by the MDRD model, in which age, gender, and ethnicity were adjusted for creatinine excretion [28]. In cancer patients, nutritional status and weight are largely affected by the disease status and side effects from medical and surgical treatments. These factors could interfere with the measurement of the actual creatinine excretion rate. By adjusting for weight in addition to the other three factors used in the MDRD model, it is possible that the CKD-EPI model has a better performance than the other two, despite the concerns about the validity of the MDRD and CKD-EPI models for the Asian population [29,30]. However, in clinical practice, results from both the MDRD and CKD-EPI models might not be readily available and thus limit the generalization of their use.

The Youden index for the UPCR was 1.83. To facilitate clinical practice, we calculated the cut-off at 2.0 for the UPCR as well. In further analysis, the simple UD, UACRD, and UPCR tests had good sensitivity and thus could rule out ≥ 2 g/day. Meanwhile, the UPCR 1.83 and 2.00 had good specificity (86% and 88%, respectively). On the contrary, the specificity for the dipstick tests was low, where the false positive rates were high (72% for the simple UD test and 83% for the UACR dipstick test). Furthermore, the price for the dipstick tests and the 24-h urine total protein content for confirmation was 14.53% higher than the price for the UPCR tests. Despite the small number of cases at a nephrotic range of proteinuria, the UPCR 2.00 had 100% sensitivity. Previously, one study suggested that the routine use of the UPCR in bevacizumab-treated patients with gynecologic malignancies for the detection of grade 3 (≥3.5 g/day) or more proteinuria is not recommended because of the low prevalence of the disease and high cost for the UPCR test [6]. In that study, proteinuria grade ≥ 3 was only observed in 2% of patients. Our previous study also observed that the prevalence of grade ≥ 3 proteinuria plateaued around 3% after reaching a cumulated bevacizumab dose of 4530 g [7]. However, to identify those who are at risk before developing a nephrotic range of proteinuria, a lower threshold is feasible. In that study, the cost of UPCR was five times higher than the simple UD test (USD 99 and 18 per test, respectively). To reduce unnecessary tests, improve the quality of life and healthcare experience, and increase care team satisfaction, adopting a more specific screening tool for the monitoring of proteinuria in gynecologic cancer patients receiving bevacizumab should be considered [31]. For the screening of urine protein content ≥ 2 g/day, when the cost is reasonable, it is easy and effective to use the UPCR test. The 24-h urine total protein content measurement, which is more time-consuming and difficult to conduct, is reserved for patients with a UPCR result of 2 mg/mg or more.

We identified several limitations in our study. First, we retrospectively collected patients’ medical records, which could introduce potential recall bias or error in recording. Second, the 24-h urine total protein content could have been inaccurately collected since some tests were conducted in an outpatient setting. Therefore, the correlation between the 24-h urine protein content and the various urine tests might be altered. Third, all patients included in this study were of Asian ethnicity, which could alter the generalizability of our results. Fourth, the costs of these tests differ among institutions, countries, or reimbursement systems. Also, the cost analysis did not consider the patients’ and medical care providers’ physical and mental alterations. Lastly, it is still debatable whether bevacizumab treatment should be stayed or discontinued for grade 3 or higher proteinuria since the associated complications are rare.

## 5. Conclusions

The UPCR test strongly correlated with the 24-h urine total protein content for gynecologic cancer patients under bevacizumab treatment. To identify patients at risk for a nephrotic range of proteinuria, reduce false positive results, and aid in decision-making on the continuation of bevacizumab therapy, a screening test using the UPCR test with a cut-off of 2.00, followed by a 24-h urine protein content analysis for those with positive results, is easy and cost-effective, especially when the prevalence of proteinuria is low.

## Figures and Tables

**Figure 1 diagnostics-14-01852-f001:**
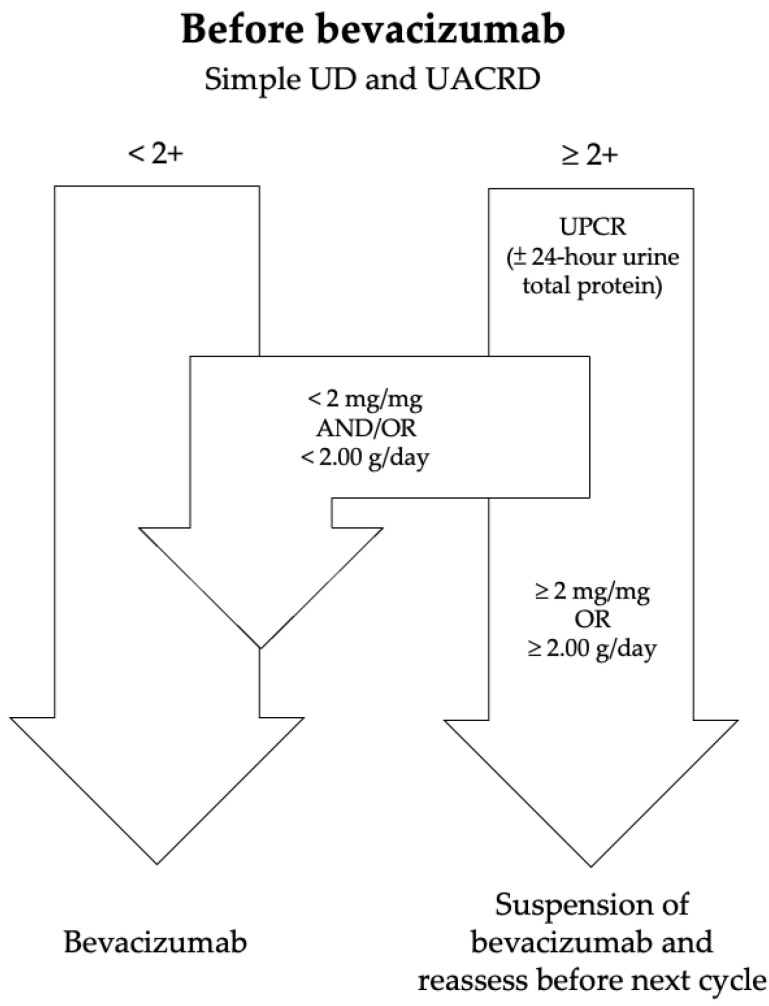
Flowchart for the screening and management of proteinuria in patients receiving bevacizumab.

**Figure 2 diagnostics-14-01852-f002:**
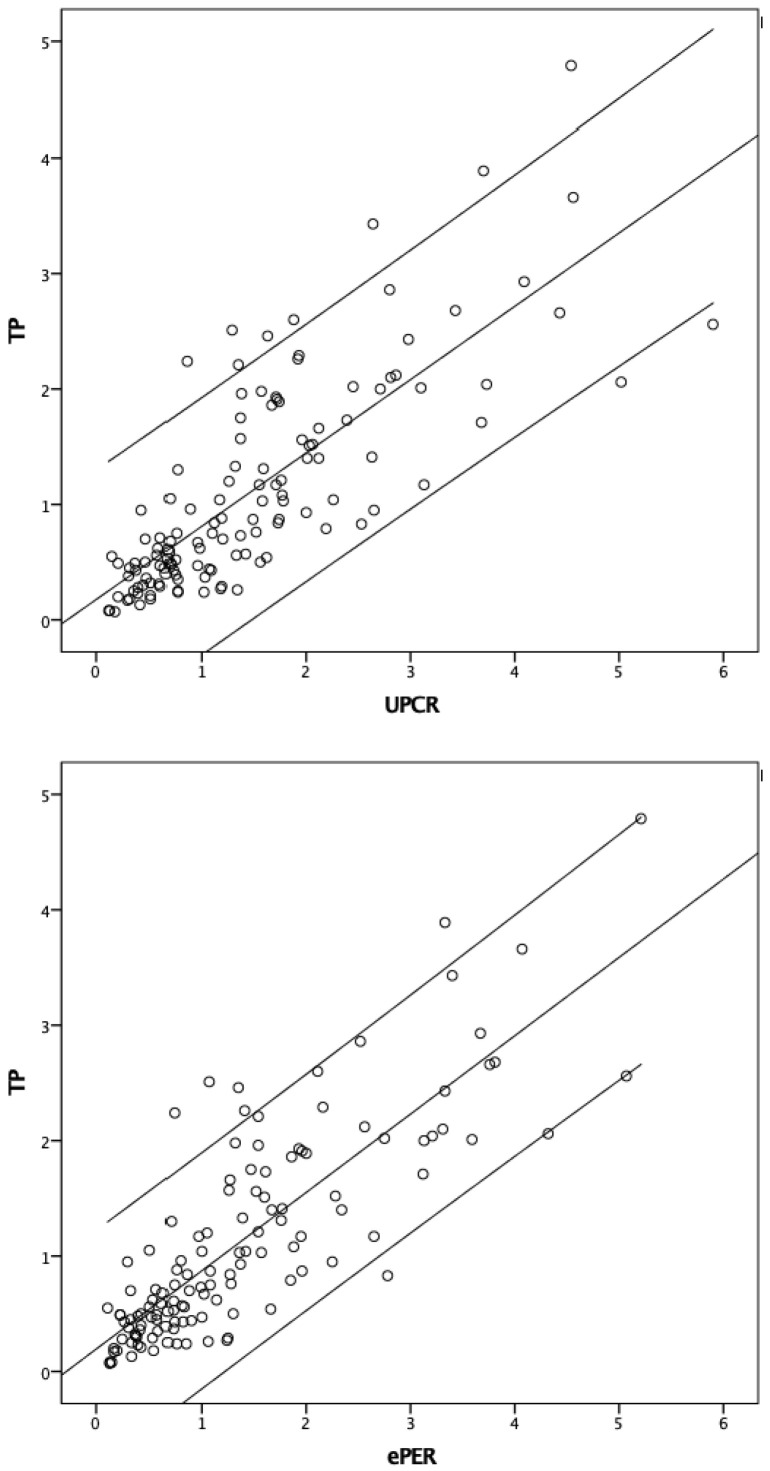
Correlation coefficiency between UPCR, ePER-CKD-EPI, and 24-h urine total protein.

**Figure 3 diagnostics-14-01852-f003:**
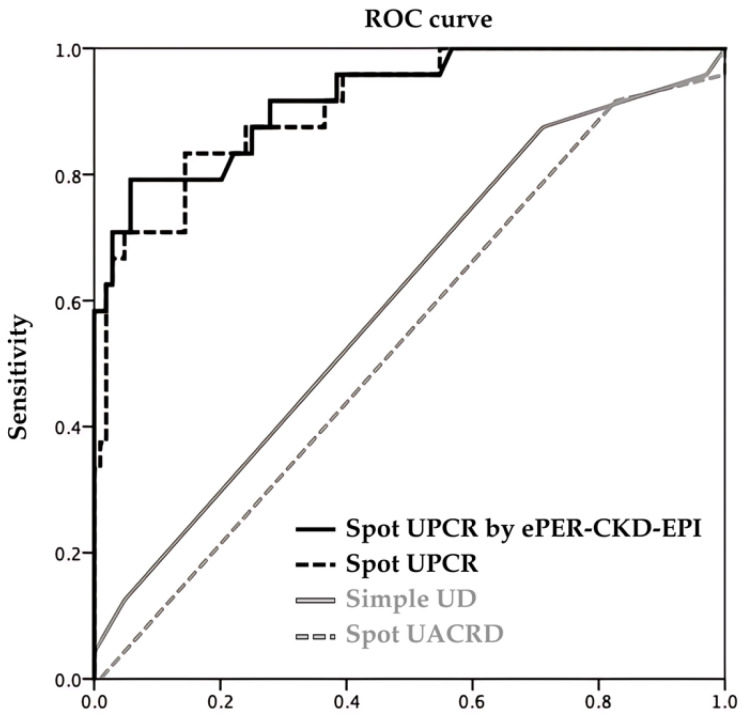
The receiver operating characteristic curve for spot urine screening tests. UPCR = urine total protein (mg/dL)/urine creatinine (mg/dL); ePER-CKD-EPI = UPCR × (879.89 + 12.51 × weight − 6.19 × Age − 379.42)/10^4^.

**Table 1 diagnostics-14-01852-t001:** Baseline characteristics.

N = 36 (129 Urine Samples)	Mean ± SD (%)
Age	63.36 ± 11.48
BMI	23.77 ± 4.39
DM	6 (16.67)
HTN	14 (38.89)
Major medical disorder(s)	25 (69.44)
Cancer origin	
Ovary, fallopian tube, and peritoneum	32 (88.89)
Uterus	2 (5.56)
Cervix	2 (5.56)
FIGO stage	
I	6 (16.67)
II	5 (13.89)
III	18 (50)
IV	7 (19.44)
Chemotherapy	
1st line	10 (27.78)
2nd line	14 (38.89)
≥3rd line	12 (33.33)
Concurrent chemotherapy	15 (41.67)
Bevacizumab dosage (mg/kg) *	162.69 ± 109.40
Proteinuria *	
<2 g/day	104 (80.62)
≥2 g/day	25 (19.38)
Percent of urine microalbumin (%) *	59.33 ± 18.94

* Calculated from 129 samples.

**Table 2 diagnostics-14-01852-t002:** Correlations between random urine tests and 24-h urine total protein.

Item	r	95% CI for r	*p*-Value	Linear Regression
Albumin				
UACRD	0.10	(0.12)–0.32	0.311	y = 0.21x + 0.76
UACR	0.75	0.67–0.82	<0.001	y = 0.84x + 0.30
UACR-MDRD	0.74	0.67–0.81	<0.001	y = 0.79x + 0.34
eAER-CKD-EPI	0.78	0.71–0.84	<0.001	y = 0.90x + 0.31
Total protein				
Simple UD	0.35	0.15–0.53	<0.001	y = 0.52x + 0.22
UPCR	0.79	0.71–0.85	<0.001	y = 0.64x + 0.18
UPCR-MDRD	0.78	0.71–0.85	<0.001	y = 0.60x + 0.22
ePER-CKD-EPI	0.82	0.75–0.88	<0.001	y = 0.68x + 0.19

UACRD: urine albumin to creatinine ratio dipstick; UACR: urine (micro)albumin to creatinine ratio; Simple UD: simple urine dipstick; UPCR: urine protein to creatinine ratio; UACR-MDRD: Modification of Diet in Renal Disease equation for the estimated glomerular filtration rate = UACR × (1051.3 + 5.3 × Age − 0.1 × Age^2^)/10^4^ for Asian female; eAER-CKD-EPI: Chronic Kidney Disease Epidemiology Collaboration equation for the estimated glomerular filtration rate in the assessment of microalbumin to creatinine ratio and 24-h urine total protein = UACR × (879.89 + 12.51 × weight − 6.19 × Age − 379.42)/10^4^ for female; ePER-CKD-EPI: Chronic Kidney Disease Epidemiology Collaboration equation for the estimated glomerular filtration rate in the assessment of protein to creatinine ratio and 24-h urine total protein.

**Table 3 diagnostics-14-01852-t003:** Spot Urine Screening Tests and Their Performance for Grade 2+ and More Proteinuria.

	UACRD	Simple UD	UPCR 1.83	UPCR 2.0	ePER 2.06
Sensitivity	0.92	0.88	0.83	0.71	0.79
Specificity	0.17	0.28	0.86	0.88	0.94
PPV	0.21	0.22	0.57	0.55	0.76
NPV	0.90	0.91	0.95	0.93	0.95
AUC	0.54	0.60	0.91	0.91	0.92
Post hoc power	15.7%	35.6%	100%	100%	100%
Cost/person (USD)	4.71	4.54	3.93	3.88	3.80

PPV: positive predictive value; NPV: negative predictive value; AUC: area under the curve; UACRD: urine albumin to creatinine ratio dipstick; UACR: urine (micro)albumin to creatinine ratio; Simple UD: simple urine dipstick; UPCR: urine protein to creatinine ratio; ePER: estimated glomerular filtration rate.

## Data Availability

Data will be shared upon reasonable request.

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
