# Peer review of "Urine Protein to Creatinine Ratio for the Assessment of Bevacizumab-Associated Proteinuria in Patients with Gynecologic Cancers: A Diagnostic and Quality Improvement Study"

_diagnostics, 2024, doi:10.3390/diagnostics14171852_

Round 1

Reviewer 1 Report

Comments and Suggestions for Authors

Thank you for the opportunity to review this manuscript. The rationale for the work is well described. The study design is clearly articulated and carried out. The data is presented well and appropriate statistics applied. The discussion is appropriate to the results and the current conduct of urine testing. Conclusions are appropriate.

Issue: Cost effectiveness analysis is pertinent to the question. This was not one of the study objectives but was raised in the methods (3.3) . The authors did not actually report on costing values, ICErs ect. This section needs ammending

Reviewer 2 Report

Comments and Suggestions for Authors

The authors put up a nice statitistic model to try to assess bevacizumab associated proteinuria. This is a known adverse reaction in patients undergoing bevacizumab treatment, especially for gynecologic malignancies, which by current standards is diagnosed through the use of 24 h proteinuria measurement. With this study, the authors aimed to improve the diagnostic quality and devised a correlation between the sensitivity and specificity of urine dipstick test, a urine albumin to creatinine ratio dipstick test and the classic 24-hour urine total protein levels. Starting from this rather simple objective, the statistical correlation results proved a lot of work in assessing even the cost efficiency. There are still some issues that need solving:

- Abstract: "microalbumin, total protein, and creatinine" urine or seric levels? please specify

- Abstract, line 37:"urine dipstick" they're all urine dipsticks. try to attach a name to it, in order to differentiate it from UACR or UPCR. e.g. simple urine dipstick or spot urine dipstick as you put it in Introduction paragraph. Additionally abstract should elaborate on the objectives of this study, mention the proposed models for the 24 h total protein content and also the cost efficiency analysis.

- Introduction, line 49: "gm" ? If you mean grams, it's just "g". The same applies to Fig 1 and throughout the text.

- Introduction, line 53-56: "Generally...nephrotic syndrome" . Please add a reference for these 2 statements

- Materials and methods, Figure 1: "withdraw" what?

- Materials and methods, line 101: "in which the albumin..." Do you mean that the UACRD measures albumin to creatinine ratio or are you reffering to previously measured seric or urine albumin? Please rephrase for better clarity.

- line 103: there should always be a space between the value and the unit of measure

- line 107: "immediately" This implies that all patients had already collected the 24 h urine right before UD or UACRD test. please specify for better clarity

- Table 1: for patients with uterus and cervix origin was there any radiotherapy involved? if so, irradiation field size and dose to kidneys should be assessed as another probable cause for kidney disfunction.

- Table 1: what's the relevance of figo staging? there is no uniform cancer origin, each figo stage includes patients with different malignancies. It's not a stringent request, just want to know your point of view.

- Table 2: the real attracting topic of this paper

Round 2

Reviewer 2 Report

Comments and Suggestions for Authors

Good work !